# Effect of the Pandemic Outbreak on ICU-Associated Infections and Antibiotic Prescription Trends in Non-COVID19 Acute Respiratory Failure Patients

**DOI:** 10.3390/jcm11237080

**Published:** 2022-11-29

**Authors:** Enrico Bussolati, Rosario Cultrera, Alessandra Quaranta, Valentina Cricca, Elisabetta Marangoni, Riccardo La Rosa, Sara Bertacchini, Alessandra Bellonzi, Riccardo Ragazzi, Carlo Alberto Volta, Savino Spadaro, Gaetano Scaramuzzo

**Affiliations:** 1Department of Translational Medicine, University of Ferrara, 44121 Ferrara, Italy; 2Infectious Diseases Unit, Azienda Ospedaliera Universitaria Sant’Anna, 44121 Ferrara, Italy; 3Intensive Care Unit, Azienda Ospedaliera Universitaria Sant’Anna, 44121 Ferrara, Italy

**Keywords:** healthcare-associated infections, multidrug resistance, COVID-19, acute respiratory failure, mechanical ventilation, antimicrobial therapy, SARS-CoV-2 pandemic1

## Abstract

Background: The COVID-19 pandemic had a relevant impact on the organization of intensive care units (ICU) and may have reduced the overall compliance with healthcare-associated infections (HAIs) prevention programs. Invasively ventilated patients are at high risk of ICU-associated infection, but there is little evidence regarding the impact of the pandemic on their occurrence in non-COVID-19 patients. Moreover, little is known of antibiotic prescription trends in the ICU during the first wave of the pandemic. The purpose of this investigation is to assess the incidence, characteristics, and risk factors for ICU-associated HAIs in a population of invasively ventilated patients affected by non-COVID-19 acute respiratory failure (ARF) admitted to the ICU in the first wave of the COVID-19 pandemic, and to evaluate the ICU antimicrobial prescription strategies. Moreover, we compared HAIs and antibiotic use to a cohort of ARF patients admitted to the ICU the year before the pandemic during the same period. Methods: this is a retrospective, single-centered cohort study conducted at S. Anna University Hospital (Ferrara, Italy). We enrolled patients admitted to the ICU for acute respiratory failure requiring invasive mechanical ventilation (MV) between February and April 2020 (intra-pandemic group, IP) and February and April 2019 (before the pandemic group, PP). We excluded patients admitted to the ICU for COVID-19 pneumonia. We recorded patients’ baseline characteristics, ICU-associated procedures and devices. Moreover, we evaluated antimicrobial therapy and classified it as prophylactic, empirical or target therapy, according to the evidence of infection at the time of prescription and to the presence of a positive culture sample. We compared the results of the two groups (PP and IP) to assess differences between the two years. Results: One hundred and twenty-eight patients were screened for inclusion and 83 patients were analyzed, 45 and 38 in the PP and I group, respectively. We found a comparable incidence of HAIs (62.2% vs. 65.8%, *p* = 0.74) and multidrug-resistant (MDR) isolations (44.4% vs. 36.8% *p*= 0.48) in the two groups. The year of ICU admission was not independently associated with an increased risk of developing HAIs (OR = 0.35, 95% CI 0.16–1.92, *p* = 0.55). The approach to antimicrobial therapy was characterized by a significant reduction in total antimicrobial use (21.4 ± 18.7 vs. 11.6 ± 9.4 days, *p* = 0.003), especially of target therapy, in the IP group. Conclusions: ICU admission for non-COVID-19 ARF during the first wave of the SARS-CoV-2 pandemic was not associated with an increased risk of ICU-associated HAIs. Nevertheless, ICU prescription of antimicrobial therapy changed and significantly decreased during the pandemic.

## 1. Introduction

Healthcare-associated infections (HAIs) and HAIs-related septic shock are the leading causes of death in noncardiac intensive care units (ICUs) and, despite advances in modern intensive care, their incidence is still rising [1]. Several factors are associated with the increase in HAIs, such as patients’ comorbidities, increased use of invasive devices, long-lasting antibiotic therapies and frequent contact with healthcare personnel caring for multiple patients [2,3,4].

The SARS-CoV-2 pandemic outbreak had an enormous impact on worldwide health, causing over 533 million confirmed cases and over 6.3 million deaths worldwide by 12 June 2022 (according to the WHO Coronavirus disease situation report). Up to 25% of infected patients were admitted to an ICU, 80% of them requiring invasive mechanical ventilation (MV) [5,6]. The magnitude of the coronavirus disease 2019 (COVID-19) pandemic required the reorganization of healthcare facilities, concerning both the increase of ICUs beds and the improvement in human and material resources. These “new” ICUs were characterized by the extensive use of personal protective equipment (PPE), increased workload and by the presence of healthcare professionals deployed from other areas [7]. All these reasons may have reduced the overall compliance with HAI prevention programs, independently of COVID-19 infection [8,9].

Although the incidence of HAIs in the COVID-19 population has been extensively studied [10,11,12,13,14,15], the indirect effect of the pandemic on the occurrence of HAIs in non-COVID-19 acute respiratory failure patients is still unknown. An association between hospitalization during the pandemic and HAIs was found in patients admitted to the neurology ward and stroke units [16], but the impact of the pandemic on HAIs in ICU non-COVID-19 ARF patients remains unknown.

Moreover, despite few data demonstrating an overall reduction in antibiotic use in outreach patients, little is known regarding the ICU antimicrobial prescription trends during the first wave of the COVID-19 pandemic.

We therefore hypothesized that the pandemic could have had indirect effects on ICU antimicrobial prescription trends and on the incidence and characteristics of ICU-associated HAIs, especially in the first wave of the pandemic.

To test this hypothesis, we assessed the incidence, characteristics and risk factors for HAIs and the ICU antimicrobial management of patients admitted to the ICU for non-COVID-19 acute respiratory failure requiring invasive mechanical ventilation during the first wave of the COVID-19 pandemic (February–April 2020). Furthermore, we compared this group to patients admitted to the same ICU during the same period in the year before the pandemic (February–April 2019).

## 2. Materials and Methods

### 2.1. Study Population and Protocol

This is a retrospective, single-center, observational cohort study of patients admitted to the ICU of the S. Anna University Hospital (Ferrara, Italy) over a period of 3 months (February, March and April) of two consecutive years, before (2019) and during the first wave (2020) of the COVID-19 pandemic. The first wave of the pandemic was defined as the time from the first detected case (31 January 2020) to the start of reopening after the national lockdown (26 April 2020). The study was approved by the institutional ethics board of Area Vasta Emilia Centrale, site in IRCCS Azienda Ospedaliera—Universitaria di Bologna, Policlinico S. Orsola-Malpighi (Protocol number 235/2022/Oss/AOUFe), and informed consent was collected or waived if collection was not possible according to the local regulations.

### 2.2. Inclusions and Exclusions Criteria

All consecutive patients admitted to the ICU during the study period were screened for inclusion. The inclusion criteria were: age 18–90 years; invasive mechanical ventilation; ICU admission for acute respiratory failure requiring invasive mechanical ventilation; and availability of a digital clinical record with detailed information on therapy and devices used during ICU stay. Exclusion criteria were: incomplete or incorrect records; unavailability of cultural samples data during ICU stay; presence of positive cultural isolations on admission and ICU admission for COVID-19-related acute respiratory failure.

### 2.3. Study Protocol and Definitions

For all patients admitted to the ICU and meeting inclusion criteria, data about demographics (i.e., age, sex, height, weight), comorbidities, ICU entrance diagnosis, medication before ICU admission, the Simplified Acute Physiology Score (SAPS) II, which is an index of disease severity [17], and duration of hospital stay before ICU admission were collected.

We collected data on ventilatory features (duration of invasive and non-invasive ventilation, oxygen therapy, tracheostomy and eventually prone positioning), invasive device features (central venous line, midline and arterial line), and presence and duration of laparostomy. Ventilatory free days (VFDs) were calculated as previously described [18]. As concerns antimicrobial therapy, we defined it as prophylactic, empiric or target according to the evidence of infection when the antimicrobial treatment was started. Specifically, we defined as (1) prophylactic any antimicrobial therapy prescribed in the absence of any sign and symptom of infection (e.g., fever, leukocytosis, increase of PCR/procalcitonin); as (2) empiric any therapy initiated without any positive cultural isolation in presence of signs and/or symptoms of infection; and as (3) target any therapy started after positive cultural isolation.

We also defined days on antimicrobial therapy as the number of days on antimicrobial treatment, independently of how many drugs were prescribed at the same time. Total antimicrobial use was defined as the cumulative sum of days on therapy for all antimicrobials during ICU stay, as previously defined by Campbell et al. [19]. Outcomes regarding length of ICU stay, mortality, microbiologic isolations (bloodstream, respiratory tract and urinary tract cultures) and multidrug resistance were also collected.

We defined HAIs as infections acquired at least 48 h after ICU admission. Bloodstream, respiratory tract and urinary tract microbial identification, antimicrobial susceptibility, multidrug resistance and MIC interpretation were defined as previously described by Cultrera et al. [10]. The isolations referring to blood, respiratory and urine cultures were requested by the attending physicians for patients with suspected secondary infections because of clinical and/or respiratory deterioration associated with suggestive laboratory or radiological findings.

### 2.4. Statistical Analyses

Categorical variables are reported as frequency, while continuous variables as mean ± standard deviation or median [interquartile range], according to data distribution (normal/not normal). Considering the absence of evidence regarding HAIs in non-COVID-19 patients during the first wave of the COVID-19 pandemic, we could not calculate a priori the sample size and therefore aimed to enroll the higher number of patients admitted to the ICU during the study period. Patients were assigned to one of the groups (PP and IP) based on the year of ICU admission.

Bivariate comparisons regarding nominal data were conducted using Pearson’s chi-square test. The Shapiro–Wilk test was used to verify continuous variables distribution. Student’s *t*-test and Mann–Whitney U test were used to compare the two samples (depending on normality distribution). Logistic regression technique was performed to evaluate risk factors associated with HAIs, and the outcome was defined as presence/absence of HAIs during ICU stay. The predictors inserted in the regression model were: the year of admission, positive history of Diabetes Mellitus, Chronic Kidney Disease (CKD), smoking, obesity (defined as BMI > 30), ICU length of stay, duration of invasive mechanical ventilation and duration of steroid therapy.

A linear multiple regression was used to test the association of SAPS II, presence of heart diseases, lung diseases, DM, CKD, year of ICU admission, duration of ICU stay, duration IMV and admission to the ICU after surgery with total antimicrobial use. Significance was set at *p* < 0.05. Statistical Analysis was performed using SPSS 24 (IBM Corp. Released 2016. IBM SPSS Statistics for Windows, Version 24.0. IBM Corp, Armonk, NY, USA) and GraphPad Prism version 8.0.0 for Windows (GraphPad Software, San Diego, CA, USA, www.graphpad.com, accessed on 1 June 2022).

## 3. Results

### 3.1. Baseline Population Characteristics

Figure 1 describes the patient selection process. Two-hundred and eleven patients were admitted to the ICU during the study period and screened for inclusion. After evaluating for inclusion and exclusion criteria, a total of 83 patients were included in the study. Their main clinical characteristics are described in Table 1.

When comparing the PP and the IP groups, the mean age of the two groups was comparable (Table 1). No significant differences were observed either in anthropometric parameters or comorbidities, with the sole exception of chronic kidney disease (*p* = 0.016). No significant differences were seen in the ICU entrance diagnosis (*p* = 0.35) and in the duration of hospital stay before ICU (*p* = 0.52, Table 1).

### 3.2. Clinical Features

The clinical characteristics during the ICU stay are resumed in Table 2. ICU length of stay (*p* = 0.92), ICU mortality (*p* = 0.68), the duration of invasive ventilation (*p* = 0.41), VFDs (*p* = 0.12) and the number of patients undergoing non-invasive ventilation, oxygen therapy, tracheostomy and pronation were not different between the two groups.

A similar number of patients between the two groups had a central venous line, a midline and/or an arterial line. In the IP group, no patient had a first central venous line inserted at the femoral site, consequently resulting in an increased number of jugular and subclavian insertion sites, although this did not reach statistical significance. The duration of steroid therapy was significantly shorter in the IP group (7.5 ± 12.3 days (PP) and 3.1 ± 5.8 (IP) (*p* = 0.038)).

### 3.3. Antimicrobial Therapy

In the IP group, the total antimicrobial use was significantly shorter (11.6 ± 9.4 days) than in the PP group (21.4 ± 18.7 days, *p* = 0.003), while the days on antimicrobial therapy were similar between the groups (6.6 ± 5.2 vs. 6.6 ± 4.2, *p* = 0.97, Table 3). The year of ICU stay was also independently associated with total antimicrobial use when adjusting for possible confounders in the regression model (Std. Beta 0.280, *p* = 0.003, Appendix A). The duration of 2nd and 3rd antimicrobials were significantly shorter in terms of days in the IP group (*p* = 0.037 and *p* = 0.019, respectively).

A higher number of patients underwent prophylactic therapy (*p* = 0.03) and a lower number of patients underwent empiric therapy (*p* = 0.05) in the PP group (Figure 2). Despite this, the number of days for each therapy was not significantly different, except for target therapy, which decreased from PP to IP (*p* = 0.03). No significative differences could be found between the groups in the antimicrobial class prescription. Nevertheless, there was a tendency towards an increased prescription of Penicillin and Carbapenems and a decreased prescription of antifungals (Figure 3).

### 3.4. Cultural Isolations

There were no significant differences between the two groups in the number of cultural samples/patient, positive cultural samples/patient and number of patients developing HAIs and MDR infections, as shown in Table 3. Microbial isolations in blood, respiratory tract and urinary tract were divided in families and differences are summarized in Appendix A. The only significant difference between PP and IP was the increased number of *Candida* spp. isolations (*p* < 0.001) in the IP group, mostly isolated from the respiratory and urinary tract.

### 3.5. Multivariate Analysis

The multivariate logistic regression analysis on risk factors associated with HAIs is shown in Table 4. In the analysis, only CKD was significantly associated with HAIs (*p* = 0.024), while being admitted to the ICU during the pandemic was not.

## 4. Discussion

In this study, ICU admission for non-COVID-19 acute respiratory failure requiring invasive mechanical ventilation during the first wave of the COVID-19 pandemic was not associated with an increased risk of healthcare-associated infection. As concerns multidrug resistance, no difference was observed in the number of patients developing MDR infections, neither considering cumulative cultures, nor respectively comparing bloodstream, respiratory tract, and urinary tract infections. Finally, we observed a change into the approach to the antimicrobial therapy, with an increased attention to antibiotic de-escalation and a lower total antimicrobial use.

Several studies explored the epidemiology of ICU infections during the COVID-19 pandemic. The overall increased incidence of HAIs reported during the pandemic [11] could be related to environmental causes (new ICU beds in other spaces in the hospital or ICU, incorporation of new doctors and nurses not previously trained in critical care, changes in the standards of patient care, use of PPI during long shifts) [12] or to the immunological and/or therapeutic characteristics of the COVID-19 infection [13]. Although HAIs in COVID-19 patients are increased [14,15], the relative role of the environmental and/or disease related factors is still not clear. By analyzing non-COVID-19 patients, we found that HAI incidence did not increase during the first wave of the pandemic. Therefore, the increased risk of HAIs already previously found in COVID-19 patients, as compared to non-COVID-19 patients [11,20], may be related to the immunological dysregulation determined by the SARS-CoV-2 virus [21] and/or to use of immunomodulatory drugs [22,23], more than it is related to environmental reasons.

Baccolini et al. [11] observed a higher proportion of HAIs in COVID-19 patients, compared to non-COVID-19 patients (admitted both before and during pandemic), but did not compare HAIs between non-COVID-19 patients admitted before and during the pandemic. They hypothesized a relation between better outcomes in non-COVID-19 patients and a less severe clinical situation on admission during the pandemic, due to social lock-down measures and fear of becoming infected inside the hospitals. Shbaklo et al. [24] observed a reduction in MDR infections during the first wave of the COVID-19 pandemic (the same period as our observation) compared with an increase in the overall bacterial infections during the late period of the pandemic. They attributed this to the growing adherence to infection prevention and control (IPC) procedures [25,26,27], suggesting that the COVID-19 pandemic may have raised awareness of the need to prevent HAIs and increased the compliance of healthcare workers to IPC in the ICU. We can confirm these findings as we found a comparable incidence of ICU HAIs before and after the start of the COVID-19 pandemic. Moreover, we found no difference in the simplified acute physiology score II (SAPS II) and in diagnosis on admission that were therefore comparable in gravity.

We also assessed the effect of the pandemic on the approach to antimicrobial therapy in ICU patients with ARF. The antimicrobial approach is determined by antimicrobial stewardship programs, listing among the objectives the sustainability of empirical and target treatments (performed through antibiotic selection), dose adjustments, drug monitoring de-escalation and shortening duration to reduce multidrug resistance and selective pressure [28]. We found that being admitted to the ICU in the before the pandemic period was independently associated with a higher risk of antimicrobial use (Appendix A). Despite this, we observed no difference in the duration of ICU stay, mortality and number of MDR infections after the shortening of both overall antimicrobial and target therapy in the IP group, as confirmed by previous evidence [29].

Our findings on the tendency to reduced antimicrobial use during the pandemic are in line with the data of the European Centre for Disease Prevention and Control (ECDC), which showed a decrease in the total antibiotic consumption in humans between 2019 and 2020 in both community [30] and hospital settings [31]. Although the report does not provide definite reasons for the reduction in antimicrobial prescription, the reasons may be found in the increase in ICU-related antimicrobial stewardship programs [32] and probably in the redistribution of resources for the ongoing pandemic, which led to a stricter tendency in antibiotic prescription. Interestingly, we also reported a decrease in the duration of steroid therapy during the first wave of the pandemic. Although the cumulative dose was not different among groups, the therapy was shorter in the IP group. This may also be connected to a higher awareness of the side effects of prolonged steroid therapy on HAIs and therefore is strongly linked to our findings on antibiotic prescription trends. Nevertheless, although the duration of steroid therapy was different, it was not associated with changes in HAI incidence. This could also be an issue considering the possible link between corticosteroid therapy and HAIs previously reported for COVID-19 patients [33].

When analyzing the microbial species associated with HAIs, it was found that *Candida* spp. was the only microorganism whose percentage of isolation increased between PP and IP, becoming the most frequently isolated family of the IP group. Fungal deaths increased during 2020–2021 compared with previous years, primarily driven by COVID-19, particularly those involving *Aspergillus* spp. and *Candida* spp. [34]. Poor data are available on non-COVID-19 patients admitted during the pandemic. Interestingly, the increase of *Candida* spp. infections did not seem to affect the duration of ICU stay, MV and mortality.

Our study has some limitations. First, it is a retrospective single-center cohort study evaluating a limited number of patients. Secondly, the classification of antimicrobial therapy was conducted a posteriori by analyzing the medical records. Thirdly, since the number of patients enrolled in our study is limited, the results must be considered exploratory. Finally, we only evaluated a limited period during the COVID-19 pandemic. Since some recommendations regarding antibiotic prescription changed over time [35], our findings refers only to the first wave of the pandemic and cannot be applied to the other periods.

## 5. Conclusions

ICU admission during the first wave of the COVID-19 pandemic for non-COVID-19 acute respiratory failure was not associated with a higher risk of developing hospital-associated infections. The first wave of the pandemic was characterized by an overall reduction in antimicrobial use in non-COVID-19 patients. Furthermore, this reduction was not related to an increase in hospital-acquired infections or to a worsening of ICU outcomes.

## Figures and Tables

**Figure 1 jcm-11-07080-f001:**
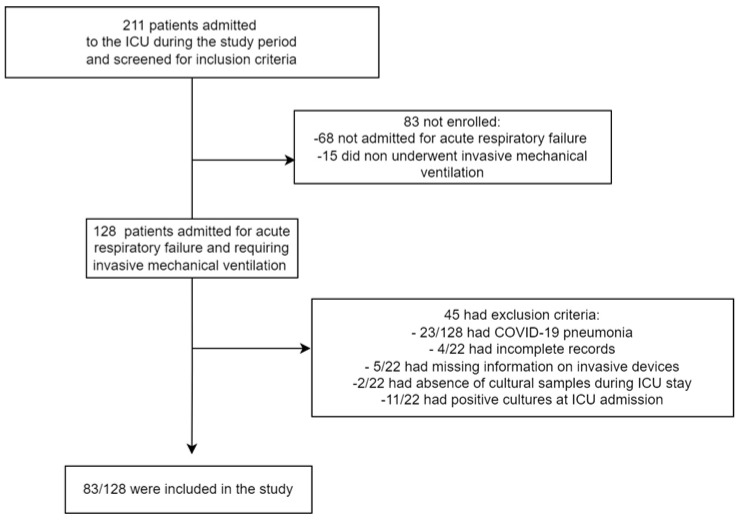
Flow diagram of patient selection process.

**Figure 2 jcm-11-07080-f002:**
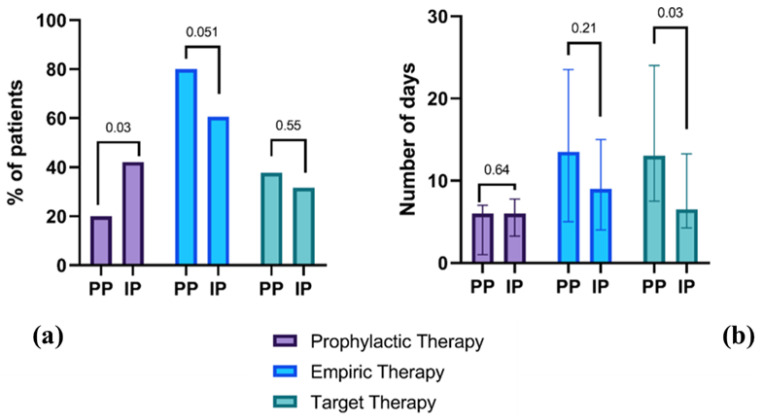
Comparison of different antimicrobial approaches before and during the pandemic. (**a**) Percentage of patients who underwent prophylactic, empiric and target therapy in the two study populations; (**b**) Number of days undergoing prophylactic, empiric and target therapy in the two years of analysis, i.e., pre-pandemic (2019) and intra-pandemic (2020). PP, pre-pandemic; IP, intra-pandemic.

**Figure 3 jcm-11-07080-f003:**
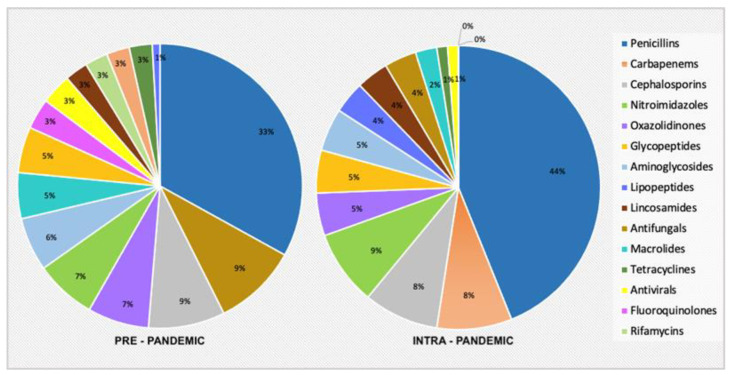
Antimicrobial classes and relative percentages regarding every administered drug in the two years of analysis (pre-Pandemic and intra-pandemic).

**Table 1 jcm-11-07080-t001:** Baseline characteristics at ICU admission, comorbidities, and entrance diagnosis.

Parameter	PP, *n* = 45	IP, *n* = 38	*p* Value
Age (years)	71.4 ± 14.1	70.16 ± 10.5	0.66
Females (number)	18 (40%)	18 (47.4%)	0.50
Weight (kg)	75.7 ± 20.1	79.8 ± 16.1	0.32
Height (cm)	168.6 ± 10	168.5 ± 7.3	0.98
BMI (kg/m2)	26.5 ± 5.5	28.1 ± 5.7	0.19
Hypertension (yes)	32 (71.1%)	28 (73.7%)	0.79
Heart Disease (yes)	25 (55.6%)	13 (34.2%)	0.052
Pneumopathy (yes)	13 (28.9%)	9 (23.7%)	0.59
CKD (yes)	13 (28.9%)	3 (7.9%)	*0.016*
DM (yes)	10 (22.2%)	11 (28.9%)	0.48
Immunosuppression (yes)	7 (15.6%)	5 (13.2%)	0.76
SAPS II	47 ± 17.7	48.1 ± 16.5	0.77
Hospital stay before ICU (days)	4.1 ± 7	6.8 ± 26.3	0.52
Smoke			0.89
*Current smokers*	8 (22.2%)	7 (18.4%)	
*Former smokers*	9 (25%)	11 (28.9%)	
Reasons for ICU amission			0.35
*Acute respiratory failure after surgery*	25 (55.6%)	19 (50%)	
*Septic Shock*	4 (8.9%)	10 (26.3%)	
*Pneumopathy*	8 (17.8%)	3 (7.9%)	
*Neuropathy*	4 (8.9%)	2 (5.3%)	
*Trauma*	2 (4.4%)	2 (5.3%)	
*Heart Disease*	1 (2.2%)	1 (2.6%)	
*Metabolic Disease*	1 (2.2%)	0 (0%)	
*Other*	0 (0%)	1 (2.6%)	

Data are expressed as Mean ± SMean SD or Number (%), according to the data. PP, pre-pandemic; IP, intra-pandemic; BMI, Body mass index; CKD, chronic kidney disease; DM, diabetes mellitus; SAPS II, Simplified Acute Physiology Score II. Italic for categories.

**Table 2 jcm-11-07080-t002:** Clinical features (outcomes, therapies, ventilatory, catheter and others) during ICU stay.

Parameter	PP, *n* = 45	IP, *n* = 38	*p* Value
ICU length of stay (days)	7.7 ± 8	7.6 ± 5.9	0.92
Dead during ICU (yes)	9 (20%)	9 (23.7%)	0.68
Duration of Invasive Ventilation (days)	4.3 ± 5.5	5.2 ± 4.7	0.41
Ventilatory Free Days (days)	24 [19.5–27]	21.5 [7.5–26.2]	0.12
Non-Invasive Ventilation (yes)	2 (4.4%)	2 (5.3%)	0.86
Oxygen therapy (yes)	34 (75.6%)	28 (73.7%)	0.84
Tracheostomy (yes)	6 (13.3%)	2 (5.3%)	0.21
Prone positioning (yes)	1 (2.3)	1 (2.6)	0.92
*Catheter Features*			
Patients with central venous line	42 (93%)	38 (100%)	0.10
Patients with midline	2 (4.4%)	0 (0%)	0.19
Patients with arterial line	44 (97.8 %)	35 (92.1%)	0.23
Central venous lines/patient during ICU stay	1 [1,2]	1 [1,2]	0.41
Site of first CVC cannulation			0.08
*Subclavian*	2 (4.4%)	3 (7.9%)	
*Jugular*	36 (80%)	35 (92.1%)	
*Femoral*	4 (8.9%)	0 (0%)	
Presence of laparostomy (yes)	3 (6.7%)	5 (13.2%)	0.32
Duration of laparostomy (Days)	5 ± 4.6	2.6 ± 1.5	0.46
Steroid Therapy during ICU stay (nr. of patients)	30 (66.7%)	22 (57.9%)	0.41
Duration of Steroid Therapy during ICU stay (days)	7.5 ± 12.3	3.1 ± 5.8	*0.038*
Total Steroid Dosage (mg Hydrocortisone/kg)	16.3 ± 31.1	14.4 ± 38	0.81
Vasoactive drugs > 0.1 γ/Kg/min (number of patients)	26 (57.8%)	27 (71.1%)	0.21

Data are expressed as Mean ± SD, Median [IQR] or Number (%), according to the data. PP, pre-pandemic; IP, intra-pandemic; CVC, central venous catheter. Italic for categories.

**Table 3 jcm-11-07080-t003:** Antimicrobial therapy, cultural samples and infections in the study population, divided for year of admission.

Parameter	PP, *n* = 45	IP, *n* = 38	*p* Value
Total antimicrobial use (days)	21.4 ± 18.7	11.6 ± 9.4	*0.003*
Day on antimicrobial therapy (days)	6.6 ± 5.2	6.64 ± 4.2	0.97
Number of different antimicrobials/patients	2.67 ± 1.6	2.22 ± 1.2	0.16
Duration of 1st Antimicrobial (days)	6.7 ± 4.8	5.3 ± 3.7	0.13
Duration of 2nd Antimicrobial (days)	5.3 ± 5.4	3.3 ± 3.3	*0.037*
Duration of 3rd Antimicrobial (days)	4.9 ± 7.6	1.9 ± 3.3	*0.019*
Patients with HAIs	28 (62.2%)	25 (65.8%)	0.74
Cultural samples per patient	8.24 ± 7.8	7.84 ± 6.2	0.79
Positive cultural samples per patient	1 [0–2.5]	1 [0–3.2]	0.50
Patients with an MDR infection	20 (44.4%)	14 (36.8%)	0.48
MDR positive isolations/patient	0 [0–1]	0 [0–1]	0.52
Patients with MDR bloodstream infections	12 (26.7%)	8 (21.1%)	0.55
Patients with MDR respiratory infections	13 (28.9%)	7 (18.4%)	0.27
Patients with MDR urinary infections	1 (2.2%)	1 (2.6%)	0.90

Data are expressed as Mean ± SD, Median [IQR] or Number (%), according to the data. PP, pre-pandemic; IP, intra-pandemic; MDR, multidrug-resistant.

**Table 4 jcm-11-07080-t004:** Multivariate logistic regression analysis on risk factors associated with healthcare-associated infections, with outcome defined as presence/absence of HAIs during ICU stay.

	Multivariate Analysis			
	OR	Sig.	95% C.I. for OR	
			Lower	Upper
Year of admission (IP)	0.35	0.55	0.16	1.92
DM (yes)	0.65	0.53	0.17	2.50
CKD (yes)	14.40	0.024	1.43	146.26
Smoke (yes)	0.41	0.30	0.08	2.23
Former smokers (yes)	0.98	0.98	0.24	4.06
Tracheostomy (yes)	3.47	0.35	0.25	47.71
Obesity (yes)	0.50	0.31	0.13	1.95
ICU stay (days)	1.10	0.31	0.91	1.33
IMV duration (days)	1.22	0.09	0.97	1.55
Steroid therapy (days)	1.03	0.55	0.94	1.13

Reference in parenthesis; IP, Intra-Pandemic (2020); CKD, chronic kidney disease; DM, diabetes mellitus; ICU, intensive care unit; IMV, invasive mechanical ventilation.

## Data Availability

The data presented in this study are available on request from the corresponding author.

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
