# Peer review of "Effect of the Pandemic Outbreak on ICU-Associated Infections and Antibiotic Prescription Trends in Non-COVID19 Acute Respiratory Failure Patients"

_jcm, 2022, doi:10.3390/jcm11237080_

Round 1

Reviewer 1 Report

Thank you for giving me the opportunity to review this paper. It is an interesting research paper. Overall, the research is well designed and clear. However, I have some comments for the authors to address.

general comments: Some sentences are long and difficult to follow. I recommend the authors to break them into smaller sentences. Also there are few typos in the manuscript (e.g., the first 3 wave of the pandemic).

Specific comments.

1- line 48-50 (same considerations......... (NHSN)). This seems out of place, please consider revising or taking out this line.

2- please use "wave" instead of "part" for this line "especially in the first part of the pandemic"

3- Please define what Simplified Acute Physiology Score II is for the readers.

4- categorical variables are reported as number...... please change number to frequency.

5- I think you are missing values in table 4 for "smoke"

6- in table 2 and 3, does "Nr" means number? if yes, please be consistent.

7- I believe that you need to mention that single-centered is one of your limitations.

Reviewer 2 Report

A professionally written article that sheds light on an interesting issue. 

One issue that might be interested is to elaborate a bit more on the length of steroids administration (shorter in the PI group). 
